# Epidemiology of Celiac Disease in Cantabria, Spain

**DOI:** 10.3390/diagnostics15040505

**Published:** 2025-02-19

**Authors:** Alejandra Blanco-García, Marcos López-Hoyos, Juan Irure-Ventura, Pedro Muñoz-Cacho

**Affiliations:** 1 Department of Emergency, Marqués de Valdecilla University Hospital-IDIVAL, 39008 Santander, Spain; 2Immunopathology Group, Department of Immunology, Marqués de Valdecilla University Hospital-IDIVAL, 39008 Santander, Spain; marcos.lopez@scsalud.es (M.L.-H.); juan.irure@scsalud.es (J.I.-V.); 3Teaching Department of Primary Care Management, Cantabrian Health Service, IDIVAL, 39011 Santander, Spain

**Keywords:** celiac disease, prevalence, Cantabria, Spain

## Abstract

**Background/Objectives**: Celiac disease is an enteropathy caused by a systemic autoimmune process of genetic predisposition to the ingestion of gluten. It is a public health problem worldwide because there are often long delays between the onset of symptoms and diagnosis. Our main objective is to describe the prevalence of celiac disease in our community, Cantabria, located in northern Spain. We start with an analytical database, with data collected from 2007 to 2016. We describe the possible differences in terms of age, sex, and geographic areas; family aggregation; and disease-associated comorbidities. **Methods**: The data for this research were obtained from a database from the Immunology Laboratory at the Marqués de Valdecilla University Hospital (HUMV), a reference laboratory for the entire Autonomous Community of Cantabria, located in northern Spain. The data were obtained from the analytical results collected from this database from January 2007 to January 2016, within this Community of 566,898 inhabitants in 2016. The data collected in this database consist of immunological tests, HLA-DQ2 or HLA-DQ8 antigenic patterns, focused on diagnosing celiac disease in the community of Cantabria, which have been positive during this period of time. **Results**: The prevalence of celiac disease in Cantabria is 0.14%. The mean age of diagnosis was 17.92 years. A higher percentage has been observed in the female sex and in children. **Conclusions**: The present study shows that celiac disease in the community of Cantabria is underdiagnosed. It is an important fact to consider when evaluating patients with symptoms that could be related to this disease to avoid increased use of medical consultations until a diagnosis is reached, in addition to avoiding long-term complications with this disease.

## 1. Introduction

Celiac disease is an immune-mediated enteropathy caused by a permanent sensitivity to gluten (a protein present in cereals such as wheat, barley, and rye) in genetically susceptible individuals. This leads to a small intestinal villous atrophy, crypt hyperplasia, and damage to the small intestine surface epithelium [1,2].

Until the 1990s, celiac disease was considered a rare disease that affected only children. Following the appearance of diagnostic antibody tests in 1997, it has been established that this is not a rare disease but rather a public health problem worldwide. However, the global seroprevalence has been estimated at 1.4%, and the global pooled biopsy-confirmed celiac disease is 0.7% [3].

The studies carried out estimate the prevalence at the European level at around 1%, with a range between 0.5 and 1.26%. The prevalence varies depending on the geographic area (0.4% in South America vs. 0.8% in Europe) possibly due to differences in the genetics of the population, the consumption of products with gluten, the use of antibiotics, among others. It is worth highlighting the population in Western Sahara, where the prevalence is 5.6%, the place in the world with the highest prevalence of the disease [4]. Another population with a high prevalence is the southern Brazilian Mennonites in which 3.48% has been reported [5]. In Spain, the prevalence of the six studies included in the meta-analysis by Singh P. [3] ranges between 0.17% and 3.03% [6,7,8,9,10].

If we take into account the data described above, we can deduce that there are undiagnosed cases. This is because celiac disease behaves like an iceberg, with undiagnosed silent and latent cases.

It is a disease more frequent in the female sex compared to men and can be seen in childhood and in adults [3]. In addition, there are risk groups where the prevalence is higher, as in first-degree relatives of patients with celiac disease (8–10%), so in these cases the active search for the disease is justified [11].

The clinical presentation will depend on the age and response to gluten of each patient. There are a series of classic symptoms, symptoms of malabsorption, that are more observed in children: diarrhea, abdominal distension, and lack of growth. However, the disease can present with less specific abdominal symptoms such as constipation, flatulence, dyspepsia, nausea, or vomiting. In addition, there are other extraintestinal manifestations that can make the disease difficult to identify, such as irritability, depression, anxiety, migraine, tooth enamel defects, delayed menarche, infertility, recurrent abortions, and anemia iron deficiency [2].

Celiac disease is related to other diseases such as type I diabetes, which is usually diagnosed before celiac disease; autoimmune thyroiditis; dermatitis herpetiformis; primary biliary cirrhosis; Addison’s disease; Down syndrome; Turner syndrome; neurological symptoms; and psychiatric symptoms [12]. All of them are diseases with an autoimmune basis; hence, they are related and manifest together. That is why, when faced with the diagnosis of any of these diseases, screening for celiac disease would be justified.

From the 1970s to the present, advances have been discovered and published for the proper management of the disease, such as the development of serological markers for diagnosis (antigliadin antibodies, antireticulin antibodies, anendomysial antibodies, and antitransglutaminase antibodies), the discovery of the HLA DQ2/DQ8 susceptibility genes necessary to develop celiac disease, and the discovery of the tissue transglutaminase autoantigen.

The tests carried out today for the diagnosis of celiac disease are based mainly on anti-transglutaminase antibodies, as proof of screening, due to their high sensitivity and specificity. However, it should be clear that the negativity of these serological markers does not definitively exclude the diagnosis, making it necessary to resort to more advanced techniques such as genetic study when suspicions are high.

The genetic study is useful because practically all the patients are going to present a positive genetic study: HLA-DQ2 (alleles DQA1*05 and DQB1*02) or HLA DQ8 (alleles DQA1*03 and DQB1*0302). The negativity of the study makes the diagnosis unlikely. That is, it allows us to exclude celiac disease with 99% certainty. Its determination is useful in situations such as patients with suspected negative clinical and serological studies, high-risk individuals such as relatives of first degree or with associated diseases, patients on a gluten-free diet without being diagnosed by intestinal biopsy, or patients with positive serology who refused intestinal biopsy.

The definitive diagnosis will be made by duodenal biopsy, which should always be performed before removing gluten from the diet. We will thus be able to confirm the existence of pathological lesions compatible with celiac disease and establish, according to Marsh’s classification, the stage of the disease. None of the stages of duodenal lesions are specific to the disease, but they are very suggestive, so together with the rest of the studies we will reach the definitive diagnosis.

Currently, according to the latest publication of The European Society for Paediatric Gastroenterology Hepatology and Nutrition (ESPGHAN) guidelines in 2020, to omit the intestinal biopsy, it is enough to determine total IgA and see that the levels of anti-transglutaminase IgA antibodies are more than 10 times above the high limit of normal and that the anti-endomysial IgA antibodies, in a blood sample different from that obtained for the determination of anti-transglutaminase IgA, are positive, without it being necessary that there be clinical symptoms and without being necessary to verify the positivity of HLA-DQ2 or HLA-DQ8. In cases where anti-transglutaminase IgA antibodies are positive but not in an amount of more than 10 times the high limit of normal, or anti-transglutaminase IgA is negative, an intestinal biopsy is necessary to reach a diagnosis. A genetic test should be performed when there are discrepancies between serology and histology or when we suspect a false negative result in serology.

The aim of this study is to describe the prevalence of celiac disease in the Autonomous Community of Cantabria in view of the variability in prevalence that exists worldwide. The data for this study were collected in a database from the immunology laboratory of the Marqués of Valdecilla University Hospital (HUMV), a reference laboratory for the entire Autonomous Community of Cantabria, located in the north of Spain. The data showed the analytical data from January 2007 to January 2016.

In addition to describing the sociodemographic and clinical characteristics, and the most prevalent HLA antigenic patterns in patients with celiac disease. Describe the evolution of the incidence of new diagnoses in the last decade, which will let us know if doctors have become aware of the disease over the years and perform diagnostic tests on the population.

It is important to know these data in order to raise awareness among physicians and policymakers of strategies that allow an early diagnosis of this disease, improving the symptoms of patients and preventing diseases associated with CD like intestinal lymphoma, by starting a gluten-free diet early. In relation to this, many visits to the doctor due to non-specific, mild symptoms could be saved, and this could lead to significant financial savings at the health level.

## 2. Materials and Methods

### 2.1. Study Design

This is a retrospective descriptive study.

### 2.2. Participants

The data were collected in a database from the immunology laboratory of the Marqués of Valdecilla University Hospital (HUMV), a reference laboratory for the entire Autonomous Community of Cantabria, located in the north of Spain. The data obtained from this database showed the analytical data from January 2007 to January 2016, counting this community with 566,898 inhabitants until 2016. The data collected in this database, made up of 811 patients, include immunological tests with positive results that have been requested by physicians for the purpose of diagnosing celiac disease in a patient of the Autonomous Community of Cantabria. Therefore, the study subjects will be patients of any age and sex and geographical area of Cantabria, valued at their health center by their primary care doctor or pediatrician, or in the hospital by a specialist physician, and have been diagnosed with celiac disease through an immunological analysis of the blood. These patients come from people with suspected CD or relatives of said patients; that is, they are people belonging to risk groups who represent the patients diagnosed in Cantabria. To calculate the prevalence, the total population of Cantabria will be used (566,898 inhabitants), not the total number of patients evaluated (77,244 individual persons).

### 2.3. Study Variables

The main variables of the study and its measurement scale include sex (nominal), age (numeric), health area (nominal), family history (nominal), associated diseases (nominal), and antigenic pattern (nominal). The general variable of “associated diseases” consists of 2 categories: autoimmune and non-autoimmune.

### 2.4. Inclusion Criteria

The inclusion criteria are all patients who have been studied for celiac disease, and of which positive results have been obtained in the immunological analysis of blood that it has been observed that they presented IgA-antibodies against transglutaminase 2 (IgA TGA) positive, or IgG antibodies against transglutaminase 2 positive or endomysial antibodies (EMA-IgA).

### 2.5. Statistical Analysis

For the analyses of the above data, we used the SPSS v25 program (IBM Corp., Released 2017, IBM SPSS Statistics for Windows, Version 25.0, Armonk, NY, USA).

## 3. Results

### 3.1. Distribution by Age, Sex, and Healthcare Area

The database is made up of 77,244 patients, of which, 811 were diagnosed with CD (see Figure 1). Patients were positive (262 men and 549 women), whose ages fluctuated between 0 and 85 years, with a mean of 17.92 years and a median of 7 years (Figure 2).

In relation to age, 63.8% corresponded to minors or an age equal to 17 years; 25.8% between 18 and 50 years; and 10.4% to 51 years or older.

We can observe how the prevalence decreases from the age of 18. Also, we can observe an ascending wave between 30 and 40 years.

In Table 1, we can observe a decreasing prevalence from <5 years (1.06%) to >65 years (0.002%). In addition, we can see how the prevalence in women (0.18%) is double that in men (0.09%). Regarding the four health areas of our community, a similar prevalence is shown, around 0.14%.

### 3.2. Overall Prevalence

The prevalence of celiac disease in the Autonomous Community of Cantabria in 2016 was 0.14%, taking into account that in 2016 the population of the community was 566,898 inhabitants, with a reliable interval of 0.02–1.06%.

### 3.3. Prevalence in First-Degree Relatives

In Table 2 we can see that of the 811 patients studied, 55.1% of the genetic haplotypes had been performed in fathers, 55.6% in mothers, and 42.7% in siblings. We don’t have data on 44.9% of fathers, 44.4% of mothers, and 57.2% of the siblings.

Of the fathers from whom we have data, 3.4% had celiac disease, compared with the 5.1% of mothers with celiac disease, and 15.3% of siblings with celiac disease.

On the contrary, among the asymptomatic relatives with negative typing, we find 11.4% of the fathers, 11.5% of the mothers, and 21.4% of the siblings.

### 3.4. Diagnoses by Years

Figure 3 shows the evolution of the number of patients diagnosed each year. The years 2007 and 2012 are the ones that show a greater number of diagnoses, appreciating a decrease in diagnoses in the rest of the years. In the year 2012, 139 patients were diagnosed, that is, 17% of those diagnosed in this period. We have extended the study until 2024 to see the evolution, and a gradual increase in the number of diagnoses can be seen from 2016 to 2020. A small drop is observed in diagnoses from 2020 to 2021, to increase again annually until 2024. Data for 2007 are not available due to a change in the computerized registration system, data for this year are incomplete.

### 3.5. Autoimmune-Associated Clinical Manifestations

In Table 3, we can observe the associated autoimmune clinical manifestations that we have studied with: 3% autoimmune thyroiditis, 28.1% externalized dermatitis, 1.7% demonstrated type 1 diabetes mellitus, 1.2% had Down syndrome, and 2.3% showed other autoimmune diseases, including vitiligo and psoriasis without arthropathy. It should be noted that among the dermatitis studied, 2.2% were herpetiform and 25.8% atopic.

### 3.6. Non-Autoimmune-Associated Clinical Manifestations

Among the non-autoimmune-associated clinical manifestations were migraine in 7.6% and gastroesophageal reflux in 3.1% (Table 4).

### 3.7. HLA Typing

Of the 811 studied, 83.4% of the patients (676) had undergone the study genetic HLA haplotype. A total of 40.1% had a high-risk haplotype (2.5 homozygous, 2.2 homozygous or 2.2/2.5) compared to 51.9% who had a low-risk haplotype (2.5/x, 2.2/x or 8/x) (Table 5).

## 4. Discussion

The prevalence in our study is 0.14%, a figure lower than that found in studies already carried out in which a general prevalence of 1% is estimated for celiac disease. Although in the meta-analysis carried out in 2010 by Biagi [13] where estimates the prevalence at 0.62%, with variability between European countries. Furthermore, compared to other studies carried out in Spain, the figure obtained in our case is lower [9].

The average age of diagnosis observed in our study is 17.92 years, which differs from what has been described in other series [9,10]. The two peaks of most diagnoses are in children under 5 years of age (255 cases) and in the age group between 20 and 44 years (207 cases). The second age group where more diagnoses are made could be related to the first, in that they could be the parents of the children diagnosed when carrying out the genetic study of the relatives.

The distribution by sex coincides with that described in the previous literature with greater prevalence in women than in men (0.18 vs. 0.09) [9,14,15].

Regarding the four health areas of our community, a similar prevalence is shown, around 0.14%. Except in the Reinosa area, which is 0.13, justified by a smaller population. This allows us to confirm that there are no geographical differences or external factors that cause a higher prevalence in any of the health areas into which the community is divided.

In the description by years, Figure 2, a rather significant increase in diagnoses is observed in 2012. This fact could correspond to the publication of guides for the diagnosis of celiac disease from the ESPGHAN (The European Society for Pediatric Gastroenterology, Hepatology, and Nutrition), making healthcare professionals more sensitive to the disease. After 2012, we can see how the number of annual diagnoses has remained between 70 and 80. Figure 2 also shows the incidence has increased continuously from 2015 to 2024, except for the years of the COVID pandemic, from 2015 to 2024. This increase in incidence coincides with what was reported in the UK [16], the USA [17], and Spain [18] and a meta-analysis published in [19]. However, it disagrees with the decrease in incidence observed in Sweden in the period 2010–2022, where the incidence decreased at a rate of −0.75 cases/10^5^ [20], which the authors hypothesized was the importance of environmental factors in this decreasing incidence. As well, in Spain [21], a decrease in the standardized seroprevalence of 23.62/1000 in 2004–2007 compared to 12.55/1000 in 2013–2019 is observed, the authors proposing the rotavirus vaccine as a possible cause of this decrease.

The association between first-degree relatives is revealed as described in the other literature [22,23].

Regarding the associated autoimmune clinical manifestations, percentages were similar to other series. Regarding autoimmune thyroiditis, a percentage of 3%, which conforms to what was described by Chyn Lye et al. [24]. Regarding the DM1 of 1.4% compared to 1.7% described [25]. Dermatitis, including dermatitis herpetiformis that is associated with celiac disease due to sharing a genetic base, is observed at a percentage of 2.2%, and that described in other series is 2.7% [26]. Regarding atopic dermatitis, a percentage of 25.8% was observed compared to the prevalence of the general population that it is 15–30%, difficult to specify. There are studies that show the association that exists between both pathologies [27], compared to those who found no association between the same [28]. Regarding Down syndrome, we obtain a percentage of 1.2% compared to what was described in the meta-analysis by Yang Du (6%) [29] or what was described by A. Bodas Pinedo of 6.7% [30]. Regarding other autoimmune manifestations, such as vitiligo, a percentage of 1%. In the general population, a prevalence of 0.76% as is described by Kavita Gandhi [31]. Jin-Zhan Zhang describes a prevalence of vitiligo in patients with celiac disease of 1% [32], which would be consistent with what was described in our study. Regarding psoriasis without arthropathy, which has also been described in association with celiac disease by sharing an immunological etiology, the percentage obtained has been 1.4%. In the review carried out by Prakash Acharya, the paper describes between 2.16 and 1.8% [33]. A recent phenome-wide Mendelian randomization analysis has confirmed 38 comorbidities associated with celiac disease [34], in the present study not all of them have been identified, probably due to the limited sample size.

Among the most frequently observed non-autoimmune clinical manifestations among the patients included in our study were migraine with a percentage of 7.6%. According to data from the Spanish Society of Neurology, in Spain, the migraine affects 12–13% of the population, so we could not establish an association between celiac disease and migraine. Regarding reflux gastroesophageal disease that has been observed in 3.1% of patients with celiac disease studied, nor could we make a direct association, since the prevalence in the population general is 10–15% [35].

In addition to the fact that many patients remain undiagnosed, delays in diagnosis are common. There is a wide variation between the reported mean delay, from a few months up to over 10 years [36,37]. A possible strategy to reduce this delay may be to screen diseases that are known to be associated, as has been proposed by several authors [38,39]. Another possibility would be to carry out a mass screening program in asymptomatic individuals, which has been shown in the USA to have demonstrated improvements in symptoms, quality of life, and iron deficiency after a 1-year follow-up evaluation [40].

One of the limitations of the study is that the presence of other autoantibodies, such as anti-*Saccharomyces cerevisiae* [41], anti-ganglioside antibodies, or antineuronal antibodies, has not been quantified.

Lastly, in the genetic study, it was observed that almost half of the patients studied, 40.1% and 8%, express the haplotypes of high-moderate risk in homozygosis, which conditions, together with other factors, a greater probability of presenting the disease [42,43].

## 5. Conclusions

Celiac disease is an underdiagnosed disease in our community. The dissemination of results is essential for the advancement of science. That is why this is an important piece of information to be considered, to avoid an increased use of medical consultations until the diagnosis is reached, in addition to avoiding complications of this long-term illness by establishing a gluten-free diet as soon as possible. In addition, it is essential and necessary to raise awareness among health professionals of the symptoms and diagnostic tests to be performed in patients with symptoms compatible with the illness.

## Figures and Tables

**Figure 1 diagnostics-15-00505-f001:**
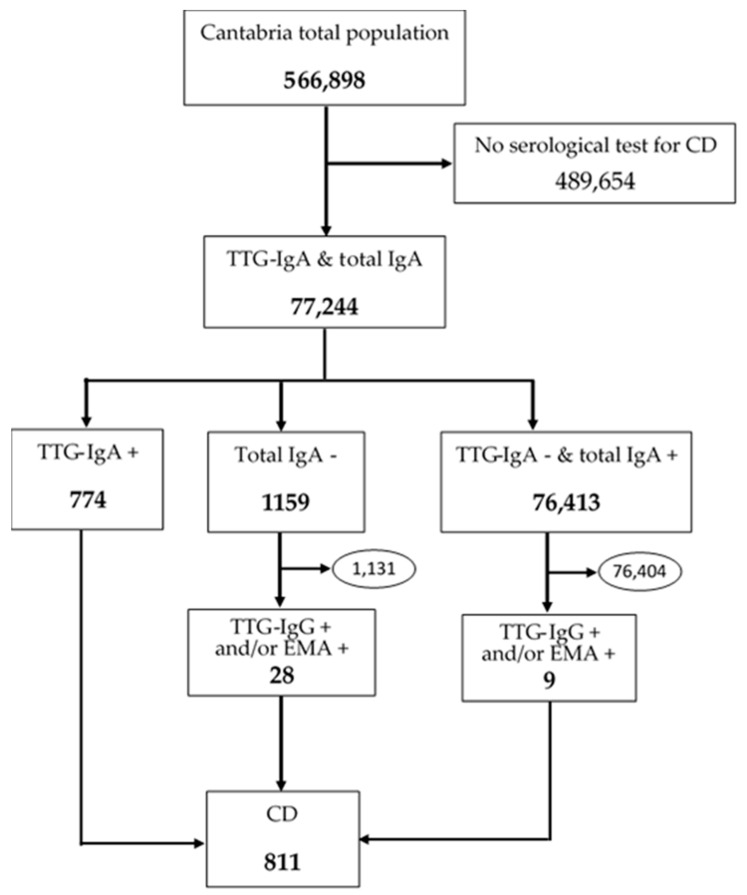
Selection process of the 811 patients diagnosed in Cantabria in the period 2007 to 2016. In this period, 77,244 people were screened for celiac disease.

**Figure 2 diagnostics-15-00505-f002:**
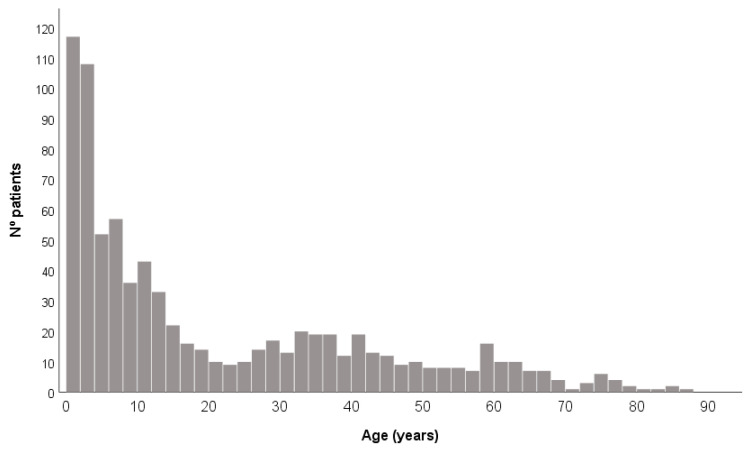
Histogram by age.

**Figure 3 diagnostics-15-00505-f003:**
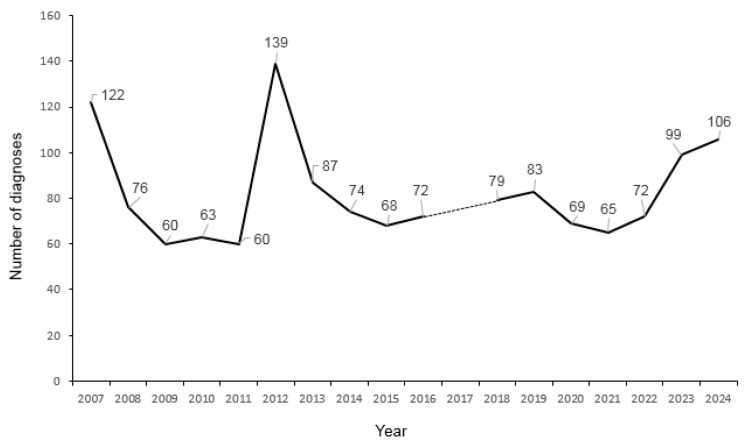
Number of diagnoses each year. The dotted line for the year 2007 is due to incomplete data due to a change in the computer system.

**Table 1 diagnostics-15-00505-t001:** Demographic data. Frequency and percentage divided into age, gender, and sanitary area in 2016.

Variable	Population Cantabria	Patients with Celiac	Prevalence (%)	95% CI
Age				
<5	24,125	255	1.06	0.94–1.20
5–9	28,284	115	0.41	0.34–0.49
10–19	50,419	128	0.25	0.21–0.30
20–44	187,152	207	0.11	0.10–0.13
45–64	171,871	85	0.05	0.04–0.06
65+	120,355	21	0.02	0.01–0.03
Total	582,206	811	0.14	0.13–0.15
Gender				
Male	282,988	262	0.09	0.08–0.10
Female	299,218	549	0.18	0.17–0.20
Sanitary Area				
I. Santander	318,889	437	0.14	0.13–0.15
II. Laredo	99,522	136	0.14	0.12–0.17
III. Reinosa	18,703	25	0.13	0.08–0.19
IV. Torrelavega	145,092	208	0.14	0.12–0.16

**Table 2 diagnostics-15-00505-t002:** First degree prevalence: frequency and percentage with data and without data in the father, in the mother, and in the brother/sister.

Variable	Father		Mother		Brother/Sister	
	N	%	N	%	N	%
With Data	447	55.1	451	55.6	346	42.7
Asymptomatic negative typing	51	11.4	52	11.5	74	21.4
Celiac	15	3.4	23	5.1	53	15.3
Asymptomatic risk typing	248	55.5	261	57.9	147	42.5
Asymptomatic without typing	133	29.8	115	25.5	72	20.8
No Data	364	44.9	360	44.4	464	57.2

**Table 3 diagnostics-15-00505-t003:** Associated autoimmune clinical manifestations: frequency and percentage of thyroiditis, dermatitis, mellitus diabetes type 1, Down syndrome, and others autoimmunes.

Variable	Frequency	%
Thyroiditis	24	3
Hiperthyroidism	9	1.1
Hipothyroidism	15	1.8
Dermatitis	227	28.1
Herpetiform	18	2.2
Atopic	209	25.8
Diabetes Mellitus Type I	14	1.7
Down Syndrome	10	1.2
Other Autoimmunes	19	2.3
Vitiligo	8	1
Psoriasis without arthropathy	11	1.4

**Table 4 diagnostics-15-00505-t004:** Non-autoimmune-associated clinical manifestations: frequency and percentage of migraine and gastroesophageal reflux.

Variable	Frequency	%
Migraine	62	7.6
Gastroesophageal Reflux	25	3.1

**Table 5 diagnostics-15-00505-t005:** HLA Typing: high risk (2.5/2.5, 2.2/2.2, and 2.2/2.5), moderate risk (8/8, 2.5/8, and 2.2/8) and low risk (2.5/x, 2.2/x, and 8/x).

Variable	Frequency	%
HLA Typing	676	83.4
High risk	271	40.1
Moderate risk	54	8
Low risk	351	51.9

## Data Availability

The raw data supporting the conclusions of this article will be made available by the authors upon request.

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
