# Peer review of "Epidemiology of Celiac Disease in Cantabria, Spain"

_diagnostics, 2025, doi:10.3390/diagnostics15040505_

Round 1
Reviewer 1 Report
Comments and Suggestions for Authors
1. It is recommended to specify in the title and abstract which country Cantabria belongs to.
2. Line 12 is missing a period. Please carefully review the entire text for similar punctuation errors.
3. Why is the literature search limited to data until 2016? Is there more recent data available?
4. Please provide a diagram illustrating the inclusion criteria for survey participants and the remaining number of participants after exclusions.
5. There is a discrepancy between the description in lines 189-192 and Table 2; should "parents" in line 189 be corrected to "father"?
6. Figure 2 and Table 3 depict the same data. Please choose one format to express this content.
7. The results section should not simply replicate the numerical data from figures and tables; instead, it should provide a summarized narrative.
Author Response
COMMENT 1: It is recommended to specify in the title and abstract which country Cantabria belongs to.
RESPONSE 1: Thank you for pointing this out. We have included in the title, in the summary (line 15) and in the section 2.2 participans (line 138), the geographical area where the study is carried out in Spain: "our community, Cantabria, located in the north of Spain". Changes are marked in red in the new version.
COMMENT 2: Line 12 is missing a period. Please carefully review the entire text for similar punctuation errors.
RESPONSE 2: Agree. We have proofread the entire text to avoid puntuaction errors.
COMMENT 3: Why is the literature search limited to data until 2016? Is there more recent data available?
RESPONSE 3: We have added bibliographic references after 2016. They have been marked in red in the reference section: reference 5 (line 422), reference 6 (line 426), reference 7 (line 429), reference 8 (line 433), reference 10 (line 439), reference 16 (line 452), reference 17 (line 456), reference 18 (line 459), reference 19 (line 462), reference 20 (line 466), reference 21 (line 468), reference 34 (line 503), reference 36 (line 508), reference 37 (line 510), reference 38 (line 517), reference 39 (line 519), reference 40 (line 522), reference 41 (line 526), reference 43 (line 532).
COMMENT 4: Please provide a diagram illustrating the inclusion criteria for survey participants and the remaining number of participants after exclusions.
RESPONSE 4: We believe that it is not neccesary to include a diagram in this study, since it is a study to estimate the prevalence of celiac disease diagnosed under normal practice conditions. In its place a comment has been added in section 2.2 participans of Material an Methods (line 147-151). If the diagram is considered essential, it can be done.
COMMENT 5: There is a discrepancy between the description in lines 189-192 and Table 2; should "parents" in line 189 be corrected to "father"?
RESPONSE 5: Thank you for pointing this out. We have corrected the description from "parents" to "father" in section 3.3 Prevalence in first degree relatives (line 210-216). The description between the text and the Table 2 now are the same.
COMMENT 6: Figure 2 and Table 3 depict the same data. Please choose one format to express this content.
RESPONSE 6: We agree with this comment. We have removed Table 3 because we believe the diagram is more graphic. For this reason we have also modified the numbering of the rest of the tables.
COMMENT 7: The results section should not simply replicate the numerical data from figures and tables; instead, it should provide a summarized narrative.
RESPONSE 7: Thank you for pointing this out. Therefor, we have made modifications in the way of describing the results in section 3.3. Prevalence in first degree relatives (line 210-216), in section 3.5. Autoimmune associated clinical manifestations (line 258-263), in section 3.6. Non-autoimmune associated clinical manifestations (line 288-289) and in section 3.7. HLA typing (line 298-300).
All changes made are marked in red.
Reviewer 2 Report
Comments and Suggestions for Authors
In this study, Blanco-García et al. aimed to assess the prevalence of celiac disease (CD) in the Community of Cantabria analyzing data from a database of the immunology laboratory, a reference laboratory from 2007 to 2016. They report that the CD prevalence in Cantabria is 0.14% with a higher rate observed in the female sex and in children underlying the need to consider CD diagnosis when evaluating patients with symptoms that could be related to this disease to avoid increased use of medical consultations, in addition to avoiding long-term complications of this disease. The study is of potential interest as epidemiological data of autoimmune diseases display geographical differerences due to a distinct genetic backgrounds. However, some important issues should be addressed.
-Study population: a critical point is the reason leading to CD serological screening. In particular, to this referee is not clear whether the 811 patients underwent CD serological screening for the presence of suggestive clinical features or other indications. Of course, if performed in high-risk patient groups (iron deficiency, malabsorption, etc) the seroprevalence might be higher than in the general population
-Introduction: Page 3, "In cases where anti-transglutaminase IgA antibodies are positive but not in an amount of more than 10 times the high limit of normal, or anti-transglutaminase IgA were negative; An intestinal biopsy should be performed to avoid errors and false diagnoses. The genetic study will be carried out in cases where a false negative serology result is suspected." This paragraph has to be modified since in patients with anti-tTG IgA <10 XUNL, intestinal biopsy is necessary (delete semicolon). A genetic test is indicated in patients with discrepancies between serology and histology due to the high negative predictive value of the genetic test. Therefore, it helps only to exclude CD diagnosis but it doesn't have satisfactory positive predictive value since about 30% of non celiac subjects are DQ2/DQ8 positive without having CD. The authors should also recall the important advancements in the immunological characterization of celiac disease patients having extraintestinal manifestations serologically marked by different circulating autoantibodies including anti-neuronal antibodies and/or anti-ganglioside antibodies in CD patients with neurological disorders, as well as the higher prevalence of CD in atopic patients (doi: 10.1016/s1590-8658(00)80354-0.).
Additionally, and of clinical relevance, the authors should recall the frequent seropositivity of celiac disease patients for anti-Saccharomyces cerevisiae (ASCA) IgG and IgA positivity potentially causing a misdiagnosis of Crohn's disease (also causing villous atrophy) as previously demonstrated (doi: 10.1111/j.1365-2036.2005.02417.x.). This is likely related to the increased mucosal permeability causing local immune response to dietary antigens.
Author Response
COMMENT 1: Study population: a critical point is the reason leading to CD serological screening. In particular, to this referee is not clear whether the 811 patients underwent CD serological screening for the presence of suggestive clinical features or other indications. Of course, if performed in high-risk patient groups (iron deficiency, malabsorption, etc) the seroprevalence might be higher than in the general population.
RESPONSE 1: Thank you for pointing this out. It is a study that represents clinical care practice in Cantabria. In fact, serological tests are performed on patients with some risk factor or relatives of patients. A sentence has been added to section 2.2 Participants of Material and Methods clarifying this. It is marked in red, line 147-151.
COMMENT 2: Introduction: Page 3, "In cases where anti-transglutaminase IgA antibodies are positive but not in an amount of more than 10 times the high limit of normal, or anti-transglutaminase IgA were negative; An intestinal biopsy should be performed to avoid errors and false diagnoses. The genetic study will be carried out in cases where a false negative serology result is suspected." This paragraph has to be modified since in patients with anti-tTG IgA <10 XUNL, intestinal biopsy is necessary (delete semicolon). A genetic test is indicated in patients with discrepancies between serology and histology due to the high negative predictive value of the genetic test. Therefore, it helps only to exclude CD diagnosis but it doesn't have satisfactory positive predictive value since about 30% of non celiac subjects are DQ2/DQ8 positive without having CD. The authors should also recall the important advancements in the immunological characterization of celiac disease patients having extraintestinal manifestations serologically marked by different circulating autoantibodies including anti-neuronal antibodies and/or anti-ganglioside antibodies in CD patients with neurological disorders, as well as the higher prevalence of CD in atopic patients (doi: 10.1016/s1590-8658(00)80354-0.).
Additionally, and of clinical relevance, the authors should recall the frequent seropositivity of celiac disease patients for anti-Saccharomyces cerevisiae (ASCA) IgG and IgA positivity potentially causing a misdiagnosis of Crohn's disease (also causing villous atrophy) as previously demonstrated (doi: 10.1111/j.1365-2036.2005.02417.x.). This is likely related to the increased mucosal permeability causing local immune response to dietary antigens.
RESPONSE 2: Agree. We have changed the paragraph according to the instructions (line 107-111).
In addition, we have added the proposed references (lines 368-370 and lines 374-376).
Reviewer 3 Report
Comments and Suggestions for Authors
**Summary:**
Celiac disease is an autoimmune disease triggered by gluten consumption, which often leads to delays in diagnosis, and represents a public health challenge worldwide. This retrospective study examines the prevalence of celiac disease in Cantabria, Spain, using data from an immunology laboratory covering the period from 2007 to 2016. The research analyzed data from approximately 567,000 residents, focusing on positive HLA-DQ2 or HLA-DQ8 tests. The results revealed a prevalence rate of approximately 0.14%, with a mean age of diagnosis of approximately 17.92 years, with a higher prevalence in women and children. The study concludes that celiac disease is underdiagnosed in Cantabria, highlighting the need for greater awareness and timely diagnosis to avoid long-term health complications. The analysis uses very basic statistical data and is supported by a bibliography that is not always international in scope.
The manuscript references are not always carefully chosen to enhance its credibility and academic rigor, in line with international academic standards. Incorporating a wide range of global studies, the references sometimes provide a solid framework that underlines the significance and relevance of the research, but at other times they rely on unclear and low-profile arguments such as Spanish-language articles.
Author Response
COMMENT 1: Celiac disease is an autoimmune disease triggered by gluten consumption, which often leads to delays in diagnosis, and represents a public health challenge worldwide. This retrospective study examines the prevalence of celiac disease in Cantabria, Spain, using data from an immunology laboratory covering the period from 2007 to 2016. The research analyzed data from approximately 567,000 residents, focusing on positive HLA-DQ2 or HLA-DQ8 tests. The results revealed a prevalence rate of approximately 0.14%, with a mean age of diagnosis of approximately 17.92 years, with a higher prevalence in women and children. The study concludes that celiac disease is underdiagnosed in Cantabria, highlighting the need for greater awareness and timely diagnosis to avoid long-term health complications. The analysis uses very basic statistical data and is supported by a bibliography that is not always international in scope.
The manuscript references are not always carefully chosen to enhance its credibility and academic rigor, in line with international academic standards. Incorporating a wide range of global studies, the references sometimes provide a solid framework that underlines the significance and relevance of the research, but at other times they rely on unclear and low-profile arguments such as Spanish-language articles.
RESPONSE 1: Thank you for pointing this out. New bibliographic references have been included, including bibliographic in spanish. All new references are marked in red in the references section.
Round 2
Reviewer 1 Report
Comments and Suggestions for Authors
Regarding to the response to my comment 3. It seems there may be a misunderstanding. I am referring to the information in line 118: "The data showed the analytical data from January 2007 to January 2016." and lines 138-140: "The data obtained from this database showed the analytical data from January 2007 to January 2016, counting this Community with 566,898 inhabitants until 2016" Why is the data limited to before 2016? Considering that nearly ten years have passed since 2016, are there any more recent data available? My concern is not about adding more references.
Regarding to the response to my comment 4.
I believe that a diagram illustrating the sample selection process would provide a clear visual representation, helping readers quickly understand the entire process from the initial data to the final selected samples. Although it may not be strictly "necessary," adding such a diagram would greatly enhance the transparency, completeness, and comprehensibility of the research methodology, and I recommend including it.
Author Response
COMMENTS 1: Regarding to the response to my comment 3. It seems there may be a misunderstanding. I am referring to the information in line 118: "The data showed the analytical data from January 2007 to January 2016." and lines 138-140: "The data obtained from this database showed the analytical data from January 2007 to January 2016, counting this Community with 566,898 inhabitants until 2016" Why is the data limited to before 2016? Considering that nearly ten years have passed since 2016, are there any more recent data available? My concern is not about adding more references.
RESPONSE 1: Thank you for the clarification. The authorization requested from the Ethics Committee was for a decade (2006-2017). However, Figure 3 shows the incidence data from 2018 to 2024, but we do not have authorization to access the clinical records, nor personal resources for their evaluation. We agree with the comment but we are unable to provide the required information.
COMMENTS 2: Regarding to the response to my comment 4. I believe that a diagram illustrating the sample selection process would provide a clear visual representation, helping readers quickly understand the entire process from the initial data to the final selected samples. Although it may not be strictly "necessary," adding such a diagram would greatly enhance the transparency, completeness, and comprehensibility of the research methodology, and I recommend including it.
RESPONSE 2: We agree. We have added the diagram that is required as Figure 1 (line 174-175). In addition, in section 2.2 Participants of Material and Methods (line 147-151) and in section 2.4 Inclusion criteria of Material and Methods (line 163-165), we have explained the selection process.
All changes made are marked in red.